# Instillation of Ophthalmic Formulation Containing Nilvadipine Nanocrystals Attenuates Lens Opacification in Shumiya Cataract Rats

**DOI:** 10.3390/pharmaceutics13121999

**Published:** 2021-11-25

**Authors:** Ryoka Goto, Shigehiro Yamada, Hiroko Otake, Yosuke Nakazawa, Mikako Oka, Naoki Yamamoto, Hiroshi Sasaki, Noriaki Nagai

**Affiliations:** 1Faculty of Pharmacy, Kindai University, Kowakae, Higashi-Osaka 577-8502, Osaka, Japan; 2133420009y@kindai.ac.jp (R.G.); 1711610066j@kindai.ac.jp (S.Y.); hotake@phar.kindai.ac.jp (H.O.); 2Faculty of Pharmacy, Keio University, Tokyo 105-8512, Japan; nakazawa-ys@pha.keio.ac.jp; 3Laboratory of Clinical Pharmacology, Yokohama University of Pharmacy, Yokohama 245-0066, Kanagawa, Japan; m.oka@hamayaku.ac.jp; 4Center for Clinical Trial and Research Support, Fujita Health University, Research Promotion and Support Headquarters, Toyoake 470-1192, Aichi, Japan; naokiy@fujita-hu.ac.jp; 5Department of Ophthalmology, Kanazawa Medical University, Kahoku 920-0293, Ishikawa, Japan; mogu@kanazawa-med.ac.jp

**Keywords:** nilvadipine, cataract, nanocrystals, ophathalmic formulation, Shumiya cataract rat

## Abstract

We developed ophthalmic formulations based on nilvadipine (NIL) nanocrystals (NIL-NP dispersions; mean particle size: 98 nm) by using bead mill treatment and investigated whether the instillation of NIL-NP dispersions delivers NIL to the lens and prevents lens opacification in hereditary cataractous Shumiya cataract rats (SCRs). Serious corneal stimulation was not detected in either human corneal epithelial cells or rats treated with NIL-NP dispersions. The NIL was directly delivered to the lens by the instillation of NIL-NP dispersions, and NIL content in the lenses of rats instilled with NIL-NP dispersions was significantly higher than that in the ophthalmic formulations based on NIL microcrystals (NIL-MP dispersions; mean particle size: 21 µm). Moreover, the supply of NIL prevented increases in Ca^2+^ content and calpain activity in the lenses of SCRs and delayed the onset of cataracts. In addition, the anti-cataract effect in the lens of rats instilled with NIL-NP dispersions was also significantly higher than that in NIL-MP dispersions. NIL-NPs could be used to prevent lens opacification.

## 1. Introduction

Cataracts lead to impaired vision via partial or complete opacification of the lens, resulting in blindness. The World Health Organization (WHO) showed that the number of visually impaired people in the world is estimated to be 314 million [1]. Although cataracts can be treated with the surgical replacement of the opacified lens, this surgery is not easily performed in developing countries. Therefore, due to a lack of access to eye care and complications associated with surgery, alternative treatments and treatment strategies using topical drugs, such as eye drops, are required to treat patients facing the threat of blindness.

Increasing age, ultraviolet light, oxidative stress, lipid peroxidation, diabetes, the oxidation of lens proteins, genetic predispositions, and various toxic agents are known as risk factors for cataract formation [2,3,4,5,6]. Among them, changes in the intracellular calcium ion (Ca^2+^) have long been known to be related to lens opacification, and the highest Ca^2+^ concentrations were related to highly localized opacities [2,3,4,5,6]. Ca^2+^ homeostasis has been reported to be important for lens transparency and structural integrity [7,8]. Moreover, increased Ca^2+^ content in the lens and the subsequent stimulation of calpain activity cause the onset of cataracts [9,10,11,12]. In addition, the regulated low intracellular Ca^2+^ level maintains the stability of the lens protein.

Nilvadipine (NIL) is a second-generation dihydropyridine calcium (Ca) channel blocker and decreases the intracellular Ca^2+^ levels by the inhibition of l-type Ca channels. The NIL was used to treat systemic hypertension and enhanced the vertebral blood flow more effectively than nicardipine and nifedipine [13]. Moreover, the l-type Ca channels, such as CaV 1.2 and 1.3 channels, are distributed and expressed in the epithelium and cortical fiber cells [14]. Intracellular Ca^2+^ was decreased by the prevention of l-type Ca channels in mouse lens epithelial cells [14]. Thus, the NIL can prevent the development of opacification and maintain lens transparency by Ca^2+^ regulation. However, traditional ophthalmic formulation (eye drops) cannot deliver an adequate drug level to the lens, and the ocular bioavailability (BA) is poor as the drugs are diluted by lacrimation and eliminate rapidly from the precorneal area by the tear turnover after instillation [15]. Therefore, delivering the required levels of NIL and maintaining the drug levels in targeting ocular tissue is important, and this challenge must be overcome for the desired pharmacological efficacy in the ophthalmic field [16].

Liposomes, dendrimers, nanocrystals, nanosuspensions, nanocarriers, microcapsules, micelles, gels, and implants are useful techniques in the ophthalmic field in terms of improving the drug delivery system (DDS), with increased ocular BA, targeted delivery, and controlled release used to overcome diffusion and penetration problems [17,18,19,20,21]. Particularly, the dispersions containing nanocrystals have been studied extensively as an ophthalmic DDS [17,18,19,20,21]. We also previously designed ophthalmic formulations based on the nanocrystals of fluorometholone, indomethacin, ketoprofen, and tranilast [22,23,24,25] and found that the instillation of dispersions containing a drug nanocrystal transfer into the intraocular field through the endocytosis pathway, such as caveolae-mediated endocytosis, clathrin-mediated endocytosis, and macropinocytosis, and the instillation of nanocrystals can enhance the ocular BA [26]. Therefore, nanotechnology based on nanocrystals may overcome the low ocular BA.

In this study, experimental animal models were used for the design of ophthalmic formulations [27,28,29]. A Shumiya cataract rat (SCR) is a hereditary cataractous rat strain [30] and one of the experimental animal models for cataracts. The opacification in an SCR lens appears in the nucleus and its peripheral areas with ageing, and the prevalence of lens opacification is expressed in 66.7% of SCRs [30]. The Ca^2+^ levels in cataractous lenses increase markedly with age compared with lenses of SCRs without cataracts, and the autolytic products of calpain are also caused in lenses of SCRs [31,32,33]. Thus, the SCR is a useful model for evaluating anti-cataract drugs. In this study, we attempted to develop ophthalmic formulations based on NIL nanocrystals (NIL-NP dispersions) and investigated whether the instillation of NIL-NP dispersions deliver the NIL to lens and prevent lens opacification in SCRs.

## 2. Materials and Methods

### 2.1. Animals

Male SCRs aged 5 weeks were housed in accordance with the Pharmacy Committee Guidelines for the Care and Use of Laboratory Animals in Yokohama University of Pharmacy and transported to Kindai University at 5 weeks of age. They were housed with controlled lighting (7:00–19:00 light; 19:00–7:00 dark) at 25 °C, unlimited access to drinking water, and a CE-2 formulation diet (Clea Japan Inc., Tokyo, Japan). Twenty microliters of NIL ophthalmic formulations (0.6% NIL) were instilled with the right eye, and all experiments were performed in accordance with the guidelines for the Association for Research in Vision and Ophthalmology (ARVO) and the Pharmacy Committee Guidelines for the Care and Use of Laboratory Animals. Moreover, the experiments using animals were approved on 1 April 2019 (project identification code, KAPS-31-003) by Kindai University.

### 2.2. Chemicals

NIL powder (NIL microcrystals, NIL-MPs), propyl p-hydroxybenzoate, Ca Test Kits, and d-mannitol were purchased from Wako Pure Chemical Industries, Ltd. (Osaka, Japan), and Cell Count Reagent SF was obtained from Nacalai Tesque Inc. (Kyoto, Japan). Benzalkonium chloride (BAC) was provided from Kanto Chemical Co., Inc. (Tokyo, Japan), and pivalephrine (0.1%) was purchased from Santen Pharmaceutical Co., Ltd. (Osaka, Japan). The Bio-Rad Protein Assay Kit was provided by Bio-Rad Laboratories (Hercules, CA, USA), and 2-hydroxypropyl-β-cyclodextrin (HPβCD) was obtained from Nihon Shokuhin Kako Co., Ltd. (Tokyo, Japan). Fluorescein was provided from Alcon Japan Ltd. (Tokyo, Japan). An LPO Assay Kit (BIOXYTECH^®^ LPO-586™) was obtained from OXIS International, Inc. (Portland, OR, USA), and the SM-4-type methylcellulose (MC) was supplied by Shin-Etsu Chemical Co., Ltd. (Tokyo, Japan). All other chemicals were of the highest purity commercially available.

### 2.3. Preparation of NIL-NP Dispersions

Dispersions containing NIL-MPs (NIL-MP dispersions) were produced by mixing NIL powder and solution (vehicle) containing MC, BAC, d-mannitol, and HPβCD, and NIL-NP dispersions were prepared following our previous reports using a Bead Smash 12 (Wakenyaku Co. Ltd., Kyoto, Japan) and a Shake Master NEO BMS-M10N21 (Bio-Medical Science Co. Ltd., Tokyo, Japan) [23,33]. Briefly, NIL-MPs were added to powder based on d-mannitol and MC and milled in an agate mortar for 60 min. Afterwards, the mixture was crushed with 1 mm zirconia beads using a Bead Smash 12 at 3000 rpm for 30 s at 4 °C. Subsequently, the mixture was transferred to a tube containing saline with HPβCD and milled with 0.1 mm zirconia beads as follows: (I) 5500 rpm, 30 s × 30 times, 4 °C, by Bead Smash 12; (II) 1500 rpm, 1 h × 3 times, 4 °C, by Shake Master NEO. The compositions of NIL-MP and NIL-NP dispersions were as follows: 0.6% NIL, 1% MC, 0.001% BAC, 0.5% d-mannitol, and 0.5% HPβCD in saline.

### 2.4. Measurement of NIL

NIL was measured by a HPLC method consisting of a LC-20AT Shimadzu pump, a DGU-20A Shimadzu degasser, an SIL-10AF auto sampler, a CTO-20A column oven, and an SPD-20A UV detector (HPLC, Shimadzu Corp., Kyoto, Japan). A quantity of 50 mM phosphate buffer/acetonitrile/methanol (50/25/25, *v*/*v*%) was applied as a mobile phase and flowed at 0.25 mL/min. An Inertsil^®^ ODS-3 column (2.1 × 50 mm) was used (GL Science Co., Inc., Tokyo, Japan), and the wavelength for detection was selected as 242 nm. In this study, 1 mg/mL propyl p-hydroxybenzoate was used as an internal standard. The measurement was performed at 35 °C using a column oven, and the samples (10 μL) were injected using a SIL-10AF. The measuring time was set at 16.5 min.

### 2.5. Characteristics of Ophthalmic Formulations Containing NIL

Characteristics of ophthalmic formulations were measured following our previous reports [34,35]. A SALD-7100 laser diffraction particle size analyzer (Shimadzu Corp., Kyoto, Japan; refractive index 1.60-0.10i) and NANOSIGHT LM10 dynamic light scattering (QuantumDesign Japan, Tokyo, Japan) were used to measure the particle size distribution, and an atomic force microscope (AFM) image of NIL nanocrystals (NIL-NPs) was provided by an SPM-9700 scanning probe microscope (Shimadzu Corp., Kyoto, Japan). The NANOSIGHT LM10 was used to measure the number of NIL-NPs. The crystalline forms of NIL-MPs and NIL-NPs were analyzed by powder X-ray diffraction (XRD) using Mini Flex II (Rigaku Co., Tokyo, Japan). In addition, the melting points of the NIL-MPs and NIL-NPs were evaluated by thermogravimetry–differential thermal analysis (TG-DTA) measurements under a nitrogen atmosphere using a simultaneous DTG-60H TG-DTA apparatus (Shimadzu Corp., Kyoto, Japan). The viscosity and zeta potential of ophthalmic formulations containing NIL were evaluated using an SV-1A (A&D Company, Limited, Tokyo, Japan) and a Model 502 Zeta Potential Meter (Nihon Rufuto Co., Ltd., Tokyo, Japan), respectively.

### 2.6. Solubility of NIL in the Ophthalmic Formulations

NIL dispersions were centrifuged at 1 × 10^5^× *g* using a MAX-XP Beckman Optima™ Ultracentrifuge (Beckman Coulter, Osaka, Japan), and soluble NIL was separated from non-solubilized NIL. Subsequently, the concentration of soluble NIL was measured by the HPLC method described above and is expressed as the solubility of NIL at 20 °C in this study [33,35].

### 2.7. Dispersibility of NIL in the Ophthalmic Formulations

Three milliliters of NIL ophthalmic formulations were added to a 5 mL test tube, and were incubated in a dark room at 20 °C. After one month, the NIL ophthalmic formulations were collected from the upper 90% of the test tube, and the concentrations of the collected samples were measured by the HPLC method described above and expressed as the dispersibility of NIL. In addition, the size distribution, the AFM image, and the number of NIL-NPs in the collected samples were measured using the NANOSIGHT LM10 and SPM-9700 described above [33,35].

### 2.8. Corneal Stimulation of NIL Ophthalmic Formulations in Cultured Human Corneal Epithelial Cells

Corneal stimulation was evaluated by human corneal epithelial cells (HCE-T cells, RIKEN BRC, Ibaraki, Japan). The 1 × 10^4^ cells of HCE-T cells were seeded in 96-well microplates (IWAKI, Chiba, Japan) and cultured in Dulbecco’s modified Eagle’s medium (DMEM, Thermo Fisher Scientific, Waltham, MA, USA) with 10 mg/L gentamicin (Wako Pure Chemical Industries, Ltd., Osaka, Japan) and 10% (*v*/*v*) heat-inactivated fetal bovine serum in humidified air containing 5% CO_2_ at 37 °C for 3 days. Subsequently, the HCE-T cells were treated for 2 min by the NIL ophthalmic formulations. Afterwards, the cells were washed with a phosphate buffer and incubated in a medium consisting of 100 µL DMEM and 10 µL Cell Count Reagent SF (Nacalai Tesque Inc., Kyoto, Japan) for 1 h, and Abs was measured at 450 nm according to the manufacturer’s instructions. The cell viability was analyzed as a ratio [Abs_treatment_/Abs_non-treatment_ (Control) × 100] [35]. In an in vivo condition, residence time in the eye is 1–2 min after instillation, due to lachrymation [36]. Therefore, we chose a treatment time of 2 min.

### 2.9. Corneal Toxicity of NIL Ophthalmic Formulations in the Rats

Seven-week-old Wistar rats were used to measure the in vivo corneal toxicity of the NIL ophthalmic formulations. Twenty microliters of NIL ophthalmic formulations (0.6% NIL) were repetitively instilled with the right eye three times a day (9:00, 15:00, and 21:00) for 2 months. The eyes were kept open for about 1 min to prevent the NIL ophthalmic formulations from being washed out. Afterwards, the 1% fluorescein was instilled to dye the wound on the cornea, which was observed using a TRC-50X Fundus camera (Topcon, Tokyo, Japan). The wound area in the image was calculated with Image J (National Institute of Health, Bethesda, MD, USA) [35].

### 2.10. Measurement of NIL Content in the Lenses

Twenty microliters of NIL ophthalmic formulations (0.6% NIL) were singly instilled with the right eye of 7-week-old Wistar rats, euthanized by injection of a lethal dose of sodium pentobarbital 0.5, 1, 2, 3, 6, and 24 h after instillation. The eyes were kept open for about 1 min to prevent the NIL ophthalmic formulations from being washed out. Afterwards, the lenses of rats were removed and homogenized in 300 µL of methanol. The homogenates were centrifuged at 20,400× *g* for 15 min at 4 °C, and the concentration of NIL in the supernatants was measured by the HPLC method described above.

### 2.11. Scheimpflug Slit Images in the SCR

Twenty microliters of NIL ophthalmic formulations (0.6% NIL) were repetitively instilled in the right eye of 6-week-old SCRs for 6 weeks (once a day), and the changes in the transparency of the lenses of these SCRs were observed by an EAS-1000 equipped with a CCD camera (Nidek, Gamagori, Japan) following a previous study [37]. Briefly, the SCRs without anesthesia were dilated by an instillation of 0.1% pivalephrine and monitored by the EAS-1000. Afterwards, the total area (pixels) of the opacity of the lenses was analyzed by the EAS-1000-equipped software. The measurement conditions were as follows: thread level, flash level, and slit length were 100 threshold level, 100 Watt-seconds, and 4.2 mm, respectively.

### 2.12. Evaluation of Cataract-Related Factors

Twenty microliters of NIL ophthalmic formulations (0.6% NIL) were repetitively instilled with the right eye of 6-week-old SCRs for 3 weeks (once a day), and these rats were euthanized by injections of a lethal dose of sodium pentobarbital 6 h after the last instillation. Afterwards, the lenses of the rats were removed and homogenized in phosphate-buffered saline (pH 7.4) on ice. The lens homogenates were centrifuged at 20,400× *g* for 30 min at 4 °C, and the supernatants were used for measurements of cararact-related factors, such as the nitric oxide (NO) level, the lipid peroxidation (LPO) level, Ca^2+^-ATPase activity, Ca^2+^ content, and calpain activity. NO levels were measured by a flow-through spectrophotometer (NOD-10, Eicom, Kyoto, Japan) [29]. The NO_2_^−^ and NO_3_^−^ were separated on a reverse-phase separation column packed with polystyrene polymer (NO-PAK, 4.6 × 50 mm, Eicom, Kyoto, Japan) NO_2_^−^ was mixed with Griess’ reagent to form a purple azo dye in a reaction coil, and the absorbance was measured at 540 nm. The amounts of NO reflect the level of NO_2_^−^ metabolites, which is produced from NO. The LPO levels were determined by measuring the lipid peroxidation products 4-hydroxynonenal and malondialdehyde using an LPO Assay Kit according to the manufacturer’s instructions and are expressed as pmol/mg of protein [37]. The Ca^2+^-ATPase activity was calculated as follows [37]: the samples were incubated in Solution A (pH 7.4, water, 100 mM, HEPES, 200 mM KCl, 2 mM EGTA, 10 mM MgCl_2_, and 2 mM ATP) with or without 2.2 mM CaCl_2_ for 1 h at 37 °C. Afterwards, the reaction was stopped by trichloroacetic acid and calculated as the difference in the Pi liberated from the ATP measured in the presence and absence of Ca^2+^ to measure the absorbance of supernatants at 660 nm. In addition, the Ca Test Kit according to the methyl xylenol blue colorimetric method was used to measure the Ca^2+^ content [38], and the Ca^2+^ content was expressed as the µmol/mg wet weight. A Calpain Activity Fluorometric Assay Kit was used as the calpain activity, and Abs (505 nm) was measured according to the manufacturer’s instructions [39]. The protein levels in samples used to determine NO levels, LPO levels, and Ca^2+^-ATPase activity were determined according to the Bradford method using a Bio-Rad Protein Assay Kit.

### 2.13. Measurement of Blood Pressure (BP)

Twenty microliters of NIL ophthalmic formulations (0.6% NIL) were repetitively instilled with the right eye of 6-week-old SCRs for 6 weeks (once a day). Systolic blood pressure (SBP) and diastolic blood pressure (DBP) were measured using a noninvasive blood pressure analysis system, BP-98A (Softron, Tokyo, Japan) [34]. The blood pressure (BP) was measured 3 h after the instillation of NIL ophthalmic formulations.

### 2.14. Statistical Analysis

Statistical analysis was performed using the Student’s t-test and ANOVA followed by Dunnett’s multiple comparison, and all values are expressed as the mean ± standard error (S.E.). *p* < 0.05 was considered a significant difference.

## 3. Results

### 3.1. Physical Properties of NIL-NP Dispersions

Figure 1 shows the size frequencies of NIL particles in the NIL ophthalmic formulations. The mean particle size of NIL without bead mill treatment was 21 ± 1.6 µm, and the size was shifted to approximately 30–150 nm by the bead mill treatment. Figure 2 shows the peak patterns of XRD and the melting point of NIL in the NIL-MP and NIL-NP dispersions. Many diffraction peaks were detected in NIL treated with or without the bead mill, and the peak patterns of NIL-MPs and NIL-NPs were similar. Moreover, the diffractogram does not change with or without bead mill treatment. These results show that milled NIL has a crystalline arrangement, and the crystalline arrangement was not changed by the bead mill treatment. Figure 3 shows the solubility, zeta potential, and viscosity of NIL in the NIL-MP and NIL-NP dispersions. The solubility of NIL was not significantly changed by the bead mill treatment, since the solubility in the NIL-NPs without HPβCD was similar to the NIL-MPs without HPβCD. The addition of HPβCD enhanced the solubility of NIL, and the solubility in the NIL-MPs with HPβCD was 2.2-fold higher than the NIL-MPs without HPβCD. On the other hand, the solubility in the NIL-NPs with HPβCD was 2.8-fold higher than the NIL-NPs without HPβCD. Thus, the constants of the HPβCD-NIL inclusion complexes in the NIL-NPs were higher than those in the NIL-MPs. The drug solubility of the NIL-NP dispersions with HPβCD was 43.6 ± 0.6 µM, and 99.7% of the NIL in the dispersions was solid. The zeta potential in the NIL-MP and NIL-NP dispersions were −69 and −74 mV, respectively, and the zeta potential in the NIL-NP dispersions was similar to that in the NIL-MP dispersions. Moreover, no difference in viscosity was observed between the NIL-MP and NIL-NP dispersions. Figure 4 shows changes in particle size frequency, nanoparticle number, concentration, and aggregation, in the NIL of the NIL-MP and NIL-NP dispersions one month after preparation. No degradation of NIL was found in the NIL-MP or NIL-NP dispersions. In the NIL-MP dispersion, precipitation was observed one month after preparation, although aggregation and precipitation was not detected in any of the NIL-NP dispersions. In addition, the nano-size was maintained in the NIL-NP dispersions, and the particle number of the NIL-NP dispersions was not changed for one month. These results show that the NIL-NP dispersions maintained dispersibility for at least one month.

### 3.2. Drug Delivery to Lens by the Instillation of NIL-NP Dispersions

Figure 5A shows the cell viability of HCE-T cells treated with NIL-MP and NIL-NP dispersions. The cell viability of the HCE-T cells treated with NIL-NP dispersions was 89.8 ± 4.8%, and no significant decrease in the cell viability of the HCE-T cells treated with the vehicle or with NIL-MP and NIL-NP dispersions was observed, in comparison with the untreated group (control). Thus, no serious cell damage was observed in HCE-T cells treated with NIL-NP dispersions. Figure 5B shows the corneal damage by the repetitive instillation of NIL-MP and NIL-NP dispersions using rats. No significant damage was observed in the groups treated with NIL-MP or NIL-NP dispersions. Over two months, no corneal wound caused by the repetitive instillation of the vehicle was observed. In addition, NIL-MP and NIL-NP dispersions also did not show any corneal wound. Figure 6 shows NIL content in the lens of rats instilled with NIL-MP and NIL-NP dispersions. The NIL was delivered to the lens by the instillation of NIL dispersions, and NIL content in the lenses of rats instilled with NIL-NP dispersions was significantly higher than that in the NIL-MP dispersions. In addition, NIL content in the right lens (instilled eye) was 9.1-fold higher than in the left lens (non-instilled eye) in the rats instilled with NIL-NP dispersions.

### 3.3. Delay Effect of NIL-NPs on the Onset of Lens Opacification in the SCR

Figure 7A,B show changes in lens opacification in the SCRs repetitively instilled with NIL-MP and NIL-NP dispersions using Scheimpflug slit images. The onset of lens opacification was started at 7 weeks of age, and mature cataracts were observed at 9 weeks of age. The repetitive instillation of NIL-MP dispersions delayed cataract development, and the onset of lens opacification started at 8 weeks of age. The repetitive instillation of NIL-NP dispersions also prevented the onset of opacification, and the preventive effect of cataract development was significantly higher than that in the non-instilled and NIL-MP-instilled SCRs. It is important to investigate whether the instillation of NIL-NP dispersions affect the BP. Therefore, we measured the changes in the BP of SCRs instilled with NIL-NP dispersions (Figure 7C). Both the SBP and DBP of SCRs repetitively instilled with NIL-NP dispersions were similar to those of non-instilled rats. Figure 8 shows the changes in cataract-related factors in the lenses of 9-week-old SCRs repetitively instilled with NIL-MP and NIL-NP dispersions. The NO levels, LPO levels, Ca^2+^ content, and calpain activity in the 9-week-old SCRs with opaque lenses were significantly higher than those in the 6-week-old SCRs with transparent lenses, and the Ca^2+^-ATPase activity was lower in 9-week-old SCRs. The NO, LPO level, and Ca^2+^-ATPase activity of the non-instilled and NIL-instilled groups were similar. On the other hand, NIL-MP and NIL-NP dispersions attenuated the increase in Ca^2+^ content and calpain activity. Moreover, the repetitive instillation of NIL-NP dispersions showed a high preventive effect in comparison with the NIL-MPs. The Ca^2+^ content and calpain activity in the lenses of SCRs repetitively instilled with NIL-NP dispersions were 0.38-fold and 0.62-fold higher than those of SCRs instilled with NIL-MP dispersions.

## 4. Discussion

The Ca^2+^ levels in cataractous lenses are higher than those in transparent lenses, and the enhanced Ca^2+^ concentration in a lens increases the calpain activity, resulting in the onset of cataract and lens opacification [9,10,11,12]. Thus, the influx and emission of tightly regulated Ca^2+^ is critical for the homeostasis of function and the physiology in lenses. In this study, we designed ophthalmic formulations containing NIL nanocrystals (NIL-NP dispersions) and showed that the instillation of NIL-NP dispersions regulated Ca^2+^ levels in the lens of SCRs, resulting in a delay of the onset of opacification and cataracts.

The selection of additives is important for the production of ophthalmic formulations based on drug nanocrystals by the bead mill method. In the ophthalmic field, BAC was used as a preservative, and a range of BAC concentrations from 0.2 to 0.001% has been applied for the preparation of commercially available ophthalmic formulations (eye drops). On the other hand, high BAC levels cause corneal damage, and the prevention of corneal damage by BAC is important to develop ophthalmic formulations with efficacy and safety. We previously reported that d-mannitol attenuated BAC toxicity without diminishing the preservative effect, and the combination of BAC and d-mannitol was safe and useful as a prescription ophthalmic formulation [40]. Therefore, we selected additives of 0.001% BAC and 0.5% d-mannitol in this study. Moreover, it has been reported that the cohesion of nanoparticulate solids was prevented by adsorption to the surface of HPβCD [33] and that MC enhances the crushing efficiency in bead mill treatment and is essential for the production of nanoparticulate solids by the bead mill method [33]. Based on these findings, we used BAC, d-mannitol, HPβCD, and MC as additives. Bead mill treatment using these additives decreased the particle size of NIL, and the size was changed to approximately 30–150 nm from 1–100 µm (Figure 1). 

It is known that the crystalline structure affects the characteristics of NIL. Therefore, we measured the crystalline structure in NIL particles treated with or without a bead mill (Figure 2). Many peaks were presented at 10–30° in both NIL-MPs and NIL-NPs, and the XRD peak patterns in the NIL-NP dispersions were similar to those in the NIL-MP dispersions. In addition, we measured the changes in the TG-DTA curve of NIL treated with or without bead mill treatment. The melting point detected at 148 °C in the NIL-MPs and the peak in the NIL-NPs were also similar. Based on these results, it was suggested that the NIL-NPs maintained a crystalline form (not amorphous), and the crystalline structure was not changed by the bead mill treatment.

Next, we determined the solubility, zeta potential, and viscosity in NIL-NP dispersions (Figure 3). We used HPβCD to prevent the cohesion of nanoparticulate solids [33]. HPβCD can form an inclusion complex with a wide variety of solid compounds, and HPβCD complexation is widely used for the solubilization of a poor solution. However, HPβCD enhanced the solubility of the NIL-MP and NIL-NP dispersions, and the ratio of the HPβCD-NIL inclusion complexes of these dispersions were different. The constants of the HPβCD-NIL inclusion complexes in the NIL-NP dispersions were higher than those in the NIL-MP dispersions. On the other hand, almost all (99.7%) of the NIL was presented as solid nanoparticles in the NIL-NP dispersions. In contrast to these results, the solubility, zeta potential, and viscosity in the NIL-NP dispersions were similar to those in the NIL-MP dispersions. We also evaluated the dispersibility of NIL in the NIL-MP and NIL-NP dispersions (Figure 4). The drug dispersibility of NIL in the NIL-NP dispersions was higher than that in the NIL-MP dispersions, and there was no observed aggregation, precipitation, or decomposition of NIL in the NIL ophthalmic formulations for one month.

The evaluation of safety and toxicity is important for ophthalmic formulations. It has been reported that BAC and HPβCD induced cell toxicity; however, it is possible to suppress these stimulations by reducing the concentration [41]. Previous reports have shown that BAC and HPβCD levels less than 0.001% and 12.5% do not cause eye irritation in the eye membrane, respectively [41]. In addition, d-mannitol attenuates BAC toxicity [40], and there is a small amount of MC toxicity in the eye [33]. In this study, cell viability in the HCE-T cells treated with the vehicle showed 0.001% BAC, 0.5% d-mannitol, 0.5% HPβCD, and 1% MC was 91.7%. Moreover, cell viability in the HCE-T cells treated with NIL-MP and NIL-NP dispersions was 88.1% and 89.8%, respectively. In addition, no corneal damage was observed in rats repetitively instilled with NIL-MP or NIL-NP dispersions. These results show that the instillation of NIL-NP dispersions is safe for ocular surfaces.

It is difficult to deliver drugs to a lens because the tear film barrier and corneal barrier restrict this kind of delivery [42]. We demonstrated whether the instillation of dispersions containing NIL nanocrystals is useful as a DDS for the lens. It is known that the drug is also detected in the non-instilled eye (left eye) when absorbed through the conjunctiva and nasolacrimal duct in the drug-instilled eye (right eye). In this study, the NIL was directly delivered to the lens by the instillation of NIL-NP dispersions, since the NIL levels in the lens of the instilled eye (right eye) were remarkable higher than those in the non-instilled eye (left eye) (Figure 6B). In addition, the NIL levels in the lens of rats instilled with NIL-NP dispersions were significantly higher than those in the NIL-MP dispersions (Figure 6A). Furthermore, the repetitive instillation of NIL-NP dispersions, in comparison with NIL-MP dispersions, attenuated the lens opacification of SCRs (Figure 7A,B). Since NIL is used to treat patients with hypertension, we measured the changes in BP after the instillation of NIL-NP dispersions. Such instillation had little effect on both SBP and DBP (Figure 7C,D). These results suggest that the ophthalmic formulations based on NIL nanocrystals are useful as a DDS for lenses and that NIL-NPs prevent cataract development in SCRs.

SCRs also serve as a useful model for revealing the mechanism of cataract development and its preventive effect in SCR lenses. The overproduction of NO can cause the oxidative inhibition of Ca^2+^-ATPase via lipid peroxidation and increased Ca^2+^ influx into the lens [43]. Afterwards, the enhanced Ca^2+^ increased calpain activity [6,44], resulting in the proteolysis of cytoskeletal proteins and crystals [45]. These biochemical mechanisms develop lens opacification in SCRs. Therefore, we investigated the effect of NIL-NP dispersions on NO levels, LPO levels, Ca^2+^-ATPase activity, Ca^2+^ content, and calpain activity in SCRs. The repetitive instillation of NIL-NP dispersions did not attenuate increases in NO or LPO levels or decreases in Ca^2+^-ATPase activity. In contrast to these results, the enhanced Ca^2+^ content and calpain activity in the lenses of SCRs were inhibited by the NIL-NP dispersions, and the preventive effect in the NIL-NP dispersions was significantly higher than that in the NIL-MP dispersions. Taken together, we hypothesize that NIL-NP dispersions directly deliver NIL to lenses and that NIL attenuates these lens’ enhanced Ca^2+^ content. Thereafter, calpain activity was inhibited by the NIL-NP dispersions, resulting in the delay of lens opacification (Figure 9). On the other hand, the Ca^2+^ and calpain activity between SCRs instilled with NIL-NP dispersions (Ca^2+^ content 15.6 ± 0.18 µmol/mg wet weight, calpain activity 187 ± 16.1%, *n* = 8) were also significantly lower than that in non-instilled SCRs (Ca^2+^ content 2.13 ± 0.19 µmol/mg wet weight, calpain activity 231 ± 10.8%, *n* = 8) in the 11-week-old SCRs with opaque lens. This result shows that the NIL have a longer effect but not strong enough to partially delay cataract formation.

The cataract can be treated with a surgical procedure, however, this surgery is not easily performed in developing countries. From this background, the delay of cataract formation will improve the patient’s sight. In addition, we previously reported that the instillation of disulfiram (radical scavenger) and lanosterol (solubilization of crystallin aggregation) also prevent the onset of cataract [39,46,47]. The combination of NIL and other anti-cataract drugs with different preventive mechanisms may be effective in treatment of cataracts, resulting in providing an impact in cataract treatments. It is important to compare the NIL-NP dispersions and other advanced functional nano eye drops [48,49], and discuss the efficient maintenance of drug levels in targeting ocular tissue and management of the diseases occurring in the inner segments of the eye. In addition, further studies are needed to evaluate the mechanism of the ocular DDS in NIL-NP dispersions. We previously reported that drug nanoparticles in ophthalmic formulations are taken into the corneal epithelium by energy-dependent endocytosis, e.g., caveolae-mediated endocytosis, clathrin-mediated endocytosis, and micropinocytosis, resulting in an enhancement of transcorneal penetration [24,26]. Therefore, we are now planning to demonstrate whether energy-dependent endocytosis, by using its inhibitors, is related to transcorneal penetration and delivery to the lens of NIL-NPs.

## 5. Conclusions

We prepared dispersions containing NIL nanocrystals and showed that the instillation of NIL-NP dispersions directly delivered NIL to the lens. Moreover, the supply of NIL by the instillation of NIL-NP dispersions delayed the progress of lens opacification in SCRs. These findings contribute to the development of therapy for cataracts.

## Figures and Tables

**Figure 1 pharmaceutics-13-01999-f001:**
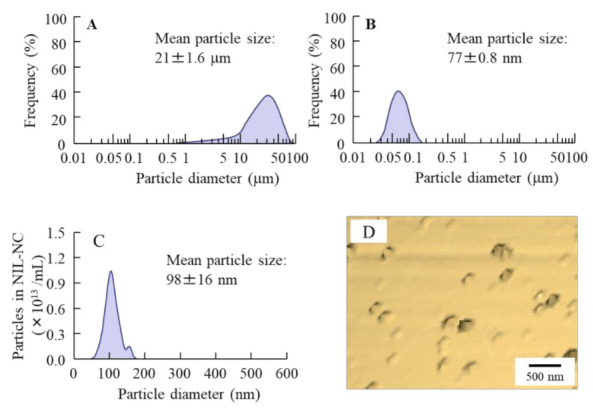
Size frequencies and AFM image of NIL particles treated with or without a bead mill. (**A**,**B**): particle size frequencies of NIL-MP (**A**) and NIL-NP (**B**) dispersions obtained by the SALD-7100. (**C**) particle size frequencies of NIL-NP dispersions obtained by the NANOSIGHT LM10. (**D**): AFM image of NIL-NP dispersions obtained by the SPM-9700. The bar shows 500 nm. The dispersions containing NIL-NPs were obtained by the bead mill treatment, and the size was approximately 30–150 nm.

**Figure 2 pharmaceutics-13-01999-f002:**
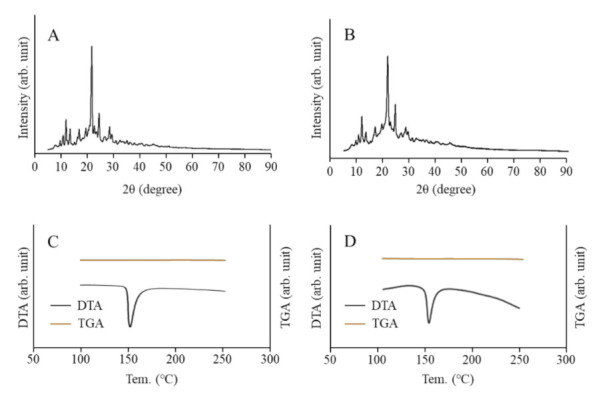
Analysis of crystalline structure and characteristics in NIL particles treated with or without a bead mill. (**A**,**B**): The powder X-ray diffraction patterns of NIL-MP (**A**) and NIL-NP (**B**) dispersions. (**C**,**D**): The TG-DTA curve of NIL-MP (**C**) and NIL-NP (**D**) dispersions. Many diffraction peaks were detected in the NIL-NPs. The peak patterns of the XRD in the NIL-NP dispersions were similar to those of the NIL-MPs, and the melting point of the NIL-NPs was not changed by the bead mill treatment.

**Figure 3 pharmaceutics-13-01999-f003:**
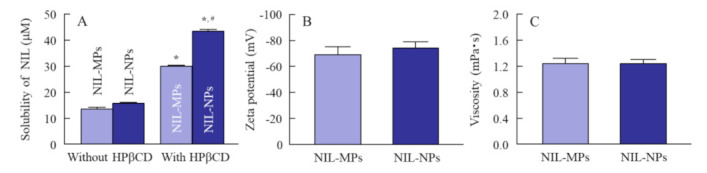
Changes in the drug solubility, zeta potential, and viscosity of the NIL dispersions treated with or without a bead mill. (**A)** Solubility in the milled NIL dispersions with or without HPβCD. (**B**) Zeta potential of the NIL in the NIL-MP and NIL-NP dispersions. (**C**) Viscosity of NIL-MP and NIL-NP dispersions. *n* = 5–8. * *p* < 0.05, vs. NIL dispersions without HPβCD for each category. ^#^ *p* < 0.05, vs. NIL-MP dispersions for each category. The solubility of NIL was increased by the bead mill treatment, and the constants of the HPβCD-NIL inclusion complexes in the NIL-NPs were also higher than those in the NIL-MPs. The zeta potential and viscosity of the NIL-MP and NIL-NP dispersions were not different.

**Figure 4 pharmaceutics-13-01999-f004:**
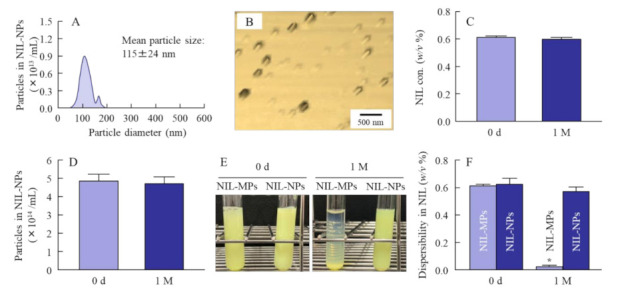
Drug stability of NIL-NP dispersions 1 month after preparation at 22 °C. (**A**) Particle size frequencies of NIL in NIL-NP dispersions. (**B**) AFM image of NIL in NIL-NP dispersions. (**C**) NIL concentration in NIL-NP dispersions. (**D**) Particle number of NIL in NIL-NP dispersions. (**E**) Representative image of NIL-MP and NIL-NP dispersions at Day 0 and 1 month after preparation. (**F**) Dispersibility of NIL-MP and NIL-NP dispersions 1 month after preparation. *n* = 6–12. * *p* < 0.05 vs. Day 0 for each category. The NIL-NP dispersions maintained dispersibility for at least 1 month.

**Figure 5 pharmaceutics-13-01999-f005:**
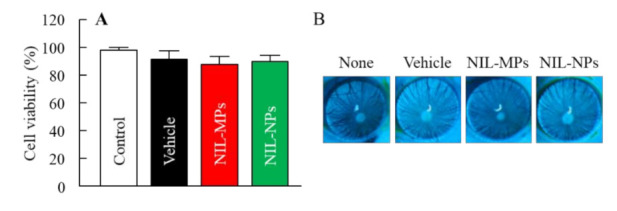
Corneal toxicity in the instillation of NIL-NP dispersions. (**A**) Effect of NIL treatment on cell viability in HCE-T cells. (**B**) Corneal image in rats repetitively instilled with NIL-NP dispersions. Twenty microliters of NIL ophthalmic formulations were repetitively instilled with the right eye three times a day for 2 months. *n* = 5. No significant damage was observed in the groups treated with NIL-MP or NIL-NP dispersions.

**Figure 6 pharmaceutics-13-01999-f006:**
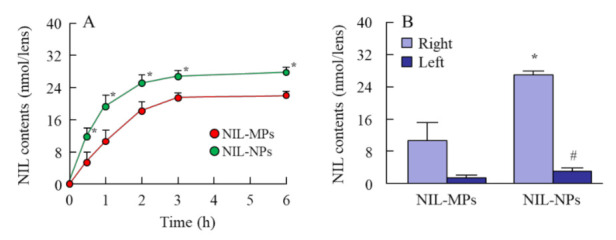
Changes in NIL content in the lens of rats instilled with NIL dispersions. (**A**) NIL content in the lens 0–6 h after the instillation of NIL dispersions. (**B**) NIL content in the lens of NIL-instilled (right) and non-NIL-instilled eye (left) 24 h after the instillation of NIL dispersions. *n* = 6–9. * *p* < 0.05 vs. NIL-MPs for each category. ^#^ *p* < 0.05 vs. the right eye for each category. The NIL levels in the lens of rats instilled with NIL-NP dispersions were significantly higher than those with the NIL-MP dispersions, and the NIL was directly delivered to the lens by the instillation of NIL-NP dispersions.

**Figure 7 pharmaceutics-13-01999-f007:**
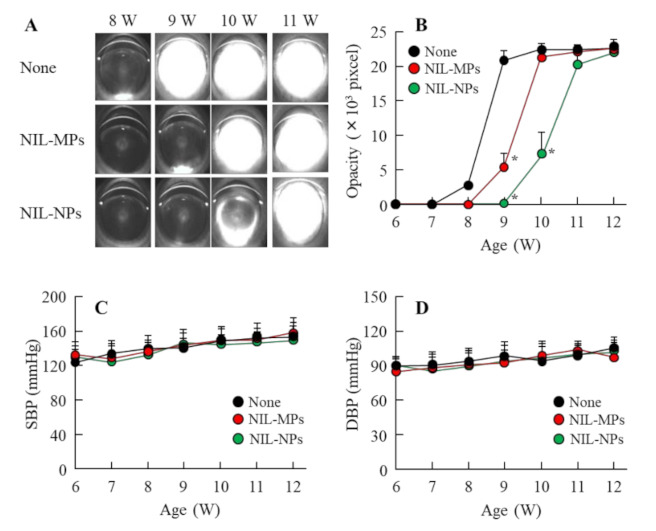
Changes in the opacity and BP in the SCRs repetitively instilled with NIL-NP dispersions. (**A**,**B**) Scheimpflug slit images (**A**) and opacity levels (**B**) in the SCRs repetitively instilled with NIL-NP dispersions. SBP (**C**) and DBP (**D**) profiles in the SCRs repetitively instilled with NIL-NP dispersions. The 6-week-old SCRs were repetitively instilled with NIL dispersions for 6 weeks (once a day). None: non-instilled SCRs. NIL-MPs: NIL-MP-instilled SCRs. NIL-NPs: NIL-NP-instilled SCRs. *n* = 4–11. * *p* < 0.05, vs. none for each group. The repetitive instillation of NIL-NP dispersions significantly delayed the onset of opacification in comparison with non-instilled and NIL-MP-instilled SCRs. On the other hand, NIL-NP dispersions did not affect the BP.

**Figure 8 pharmaceutics-13-01999-f008:**
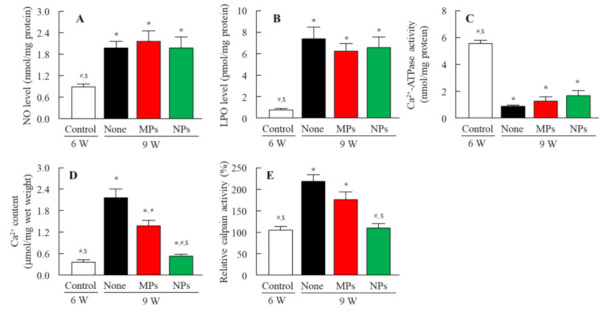
Changes in NO levels (**A**), LPO levels (**B**), Ca^2+^-ATPase activity (**C**), Ca^2+^ content (**D**), and calpain activity (**E**) in the lenses of 9-week-old SCRs repetitively instilled with NIL-NPs. The 6-week-old SCRs were repetitively instilled with NIL dispersions for 3 weeks (once a day). Control: non-instilled SCRs aged 6 weeks (transparent lens). None: non-instilled SCRs aged 9 weeks. MPs: NIL-MP-instilled SCRs aged 9 weeks. NPs: NIL-NP-instilled SCRs aged 9 weeks. *n* = 4–9. * *p* < 0.05, vs. control for each group. ^#^ *p* < 0.05, vs. none for each group. ^$^ *p* < 0.05, vs. MPs for each group. The repetitive instillation of NIL-NP dispersions significantly prevented enhanced Ca^2+^ content and calpain activity.

**Figure 9 pharmaceutics-13-01999-f009:**
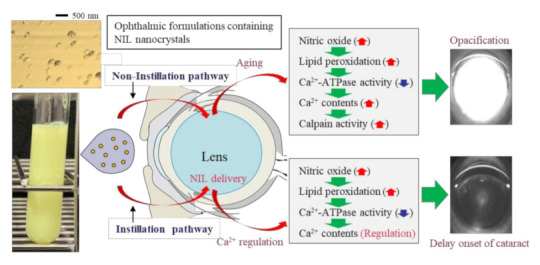
Scheme for the anti-cataract effect following the instillation of NIL-NP dispersions in SCRs.

## Data Availability

The data presented in this study are available upon request from the corresponding author.

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
