# Peer review of "Instillation of Ophthalmic Formulation Containing Nilvadipine Nanocrystals Attenuates Lens Opacification in Shumiya Cataract Rats"

_pharmaceutics, 2021, doi:10.3390/pharmaceutics13121999_

Round 1
Reviewer 1 Report
The manuscript under review reports on a study about Nilvadipine Nanocrystals preparation and characterization. The ability of the nanosuspension to prevents lens opacification in hereditary cataractous Shumiya cataract rats was also deeply investigated.
The subject of this paper is very interesting, and, in my opinion, the paper deserves publication in Pharmaceutics. However, here are a few minor points which should be amended before publication:
Line 29. It is possible that NIL-NPs can be used to prevent lens opacification. I don’t understand the meaning of this sentence. I don’t understand the meaning of this sentence. The authors have shown that NIL-NPs delayed the progress of lens opacification in SCRs. Maybe it would be better a sentence like ….. NIL-NPs could be used to prevent lens opacification.
Line 53-54. The NIL was used to treat systemic hypertension and enhanced the vertebral blood flow more effectively than nicardipine and NIL. Please check the meaning of this sentence.
Line 165-166. Three milliliters of NIL ophthalmic formulations were added to a 5 mL test tube and were incubated in a dark room at 20 °C and from the upper 90% of the test tube over one month. Please check the meaning of this sentence.
Author Response
We carefully revised our manuscript according to the suggestions of the reviewer 1, and details are as follows.
< Q and A for Reviewer 1>
Q1. Line 29. It is possible that NIL-NPs can be used to prevent lens opacification. I don’t understand the meaning of this sentence. The authors have shown that NIL-NPs delayed the progress of lens opacification in SCRs. Maybe it would be better a sentence like ….. NIL-NPs could be used to prevent lens opacification.
A1. The reviewer’s comment is correct. In order to respond to the reviewer’s comment, we revised to “NIL-NPs could be used to prevent lens opacification” (line 32).
Q2. Line 53-54. The NIL was used to treat systemic hypertension and enhanced the vertebral blood flow more effectively than nicardipine and NIL. Please check the meaning of this sentence.
A2. The reviewer’s comments are very important. We corrected to “The NIL was used to treat systemic hypertension and enhanced the vertebral blood flow more effectively than nicardipine and nifedipine”. Thank you for pointing out this. (line 55-56).
Q3. Line 165-166. Three milliliters of NIL ophthalmic formulations were added to a 5 mL test tube and were incubated in a dark room at 20 °C and from the upper 90% of the test tube over one month. Please check the meaning of this sentence.
A3. Thank you very much for pointing this out. In order to respond to the reviewer’s comment, we revised this sentence (line 168-169).
Thank you for great comments.

Reviewer 2 Report
The authors have developed topical drops of nilvadipine nanocrystals and have provided evidence that lens opacification in hereditary cataractous Shumiya cataract rats could be reduced and the onset of cataract delayed. The effect was stronger when compared to ophthalmic formulations with nilvadipine microcrystals. In their experimental Cataract rat model an increase in Ca2+ content and calpain activity was prevented and no corneal toxicity was detected in vitro and in vivo.
The topic and the findings of the manuscript are novel and are of clinical interest. The manuscript is structured, well written and understandable.
The authors showed convincing in vitro and in vivo data that nilvadipine nanocrystals reduced the severity of Cataract and delayed its onset without corneal toxicity.
Author Response
We carefully revised our manuscript according to the suggestions of the reviewer 2, and details are as follows.
< Q and A for Reviewer 2>
Thank you for great comments. We appreciate your advises.

Reviewer 3 Report
This manuscript aims to report that the instillation of ophthalmic formulation containing nilvadipine nanocrystals attenuates lens opacification in Shumiya cataract rats. Actually, the authors have previously reported a highly similar paper that already developed the same ophthalmic formulation containing nilvadipine (Deguchi et al. Ophthalmic formulation containing nilvadipine nanoparticles prevents retinal dysfunction in rats injected with streptozotocin. Int J Mol Sci 2017;18:2720.). The only difference between the current work and their earlier report is the application of this pharmaceutical formulation. From the professional viewpoint of pharmaceutics, the formulation described here is insufficient to distinguish from the previous one. Furthermore, the intention of this work is unclear since the authors have reported many different ophthalmic formulations to rescue lens structure collapse and to attenuate lens opacification in Shumiya cataract rats (Nagai et al. The intravitreal injection of lanosterol nanoparticles rescues lens structure collapse at an early stage in Shumiya cataract rats. Int J Mol Sci 2020;21:1048. & Nagai et al. Ophthalmic in situ gelling system containing lanosterol nanoparticles delays collapse of lens structure in Shumiya cataract rats. Pharmaceutics 2020;12:629.). The overall scientific progress is quite limited by insufficient academic novelty of pharmaceutical science and design. In fact, the lack of reasonable research motivation indeed constitutes one major drawback of this work. The audiences are unaware of the necessity of using the developed nilvadipine-based ophthalmic formulation again here. The pharmaceutical science/research community seems not getting any substantial new messages from this article. The authors should better clarify the novelty issue.
Specific comments:
- The group labels shown in Figure 3A are incorrect. Please carefully check the data interpretation again.
- Please clarify the data expression of y-axis shown in Figure 4D. The audiences are unaware of “IRB-NPs”.
- As stated by the authors, NIL content in the right lens (instilled eye) was 9.1-fold higher than in the left lens (non-instilled eye) in the rats (Figure 6B). However, the authors do not explain why the NIL content can be detected in the left lens (non-instilled eye)? The data should be carefully checked again.
- Why the authors used the 6-week-old SCRs as controls in Figure 8? Whether the scientific comparisons between different aged animals are reasonably acceptable?
- As stated by the authors, delivering the required levels of NIL and maintaining the drug levels in targeting ocular tissue is important. Actually, other investigators have also reported the development of advanced functional nano eye drops toward efficient maintenance of drug levels in targeting ocular tissue and management of the diseases occurred in the inner segments of the eye. Please see the recent example papers: Biomaterials 2020;243:119961 & Theranostics 2021;11:5447-5463. To attract more attention from the scholars working in the same field of ocular nanomedicine for broader applications to treat diverse eye diseases, the authors are highly recommended to consider the inclusion of these relevant publications in the reference list to enrich the article content and to balance the scientific viewpoints.
Author Response
We carefully revised our manuscript according to the suggestions of the reviewer 3, and details are as follows.
< Q and A for Reviewer 3>
Q1. The authors have previously reported a highly similar paper that already developed the same ophthalmic formulation containing nilvadipine (Deguchi et al. Ophthalmic formulation containing nilvadipine nanoparticles prevents retinal dysfunction in rats injected with streptozotocin. Int J Mol Sci 2017;18:2720.). The only difference between the current work and their earlier report is the application of this pharmaceutical formulation. From the professional viewpoint of pharmaceutics, the formulation described here is insufficient to distinguish from the previous one. Furthermore, the intention of this work is unclear since the authors have reported many different ophthalmic formulations to rescue lens structure collapse and to attenuate lens opacification in Shumiya cataract rats (Nagai et al. The intravitreal injection of lanosterol nanoparticles rescues lens structure collapse at an early stage in Shumiya cataract rats. Int J Mol Sci 2020;21:1048. & Nagai et al. Ophthalmic in situ gelling system containing lanosterol nanoparticles delays collapse of lens structure in Shumiya cataract rats. Pharmaceutics 2020;12:629.). The authors should better clarify the novelty issue.
A1. Thank you very much for pointing this out. There have been no reports for lens delivery of NIL by nanocrystalline dispersions. In addition, the result that NIL attenuate the onset of cataract is the first finding in the world. In addition, we previously reported that the instillation of disulfiram (radical scavenger) and lanosterol (solubilization of crystallin aggregation) were also prevent the onset of cataract [46,47]. The combination of NIL and these anti-cataract drugs with different preventive mechanisms may be effective in treatment of cataracts. Thus, these results show the enough novelty, and contribute to the design of anti-cataract drugs. In order to respond to the reviewer’s comment, we added these contents in the Discussion (line 489-493, References 46 and 47).
Q2. The group labels shown in Figure 3A are incorrect. Please carefully check the data interpretation again.
A2. The reviewer’s comment is correct. We corrected the group labels in Fig. 3A (Figure 3A).
Q3. Please clarify the data expression of y-axis shown in Figure 4D. The audiences are unaware of “IRB-NPs”.
A3. In order to respond to the reviewer’s comment, we collected to “NIL-NPs”. Thank you for pointing out this (Figure 4D).
Q4. As stated by the authors, NIL content in the right lens (instilled eye) was 9.1-fold higher than in the left lens (non-instilled eye) in the rats (Figure 6B). However, the authors do not explain why the NIL content can be detected in the left lens (non-instilled eye)? The data should be carefully checked again.
A4. Thank you very much for pointing this out. It is known that the drug is also detected in the non-instilled eye (left eye) when drug is absorbed through the conjunctiva and nasolacrimal duct in the drug-instilled eye (right eye). In this study, the NIL contents in the right eye with instillation was significantly higher than that in the left eye without instillation. These results suggested that the drug was delivered locally by the NIL-NP dispersions. In order to respond to the reviewer’s comment, we added these contents in the Discussion (line 450-453).
Q5. Why the authors used the 6-week-old SCRs as controls in Figure 8? Whether the scientific comparisons between different aged animals are reasonably acceptable?
A5. The reviewer’s comments are very important. The opacification was observed in the lens of 9 weeks-old SCRs. Therefore, we used the 6-week-old SCRs with transparent lens as control. In addition, we measured the NO levels, LPO levels, Ca2+-ATPase activity, Ca2+ content, and calpain activity in the lenses of 9-week-old SCRs without NIL instillation, and showed as “none” in the Fig. 8. Thank you very much for pointing this out (Figure 8 legend).
Q6. As stated by the authors, delivering the required levels of NIL and maintaining the drug levels in targeting ocular tissue is important. Actually, other investigators have also reported the development of advanced functional nano eye drops toward efficient maintenance of drug levels in targeting ocular tissue and management of the diseases occurred in the inner segments of the eye. Please see the recent example papers: Biomaterials 2020;243:119961 & Theranostics 2021;11:5447-5463. To attract more attention from the scholars working in the same field of ocular nanomedicine for broader applications to treat diverse eye diseases, the authors are highly recommended to consider the inclusion of these relevant publications in the reference list to enrich the article content and to balance the scientific viewpoints.
A6. The reviewer’s comment is correct. In order to respond to the reviewer’s comment, we added the reference and discussed in the Discussion (line 493-496, references 48 and 49).
Thank you for great comments.

Reviewer 4 Report
Dear authors,
The authors reported the instillation of Nilvadipine crystals (NIL) on rat corneas, which develop naturally cataract. The authors showed a long term stability of the NIL-MP and NIL-NP. In addition, NIL-MP and NIL-NP doesn't present a toxicity over the cornea, after 2 months of treatment. On the Shumiya cataract rat model, the installation of the NIL-MP or NIL-NP delayed the formation of the cataracts by 2 weeks and 1 week respectively. The manuscript is well written and easy to understand.
I have some comments:
- In the abstract, the authors should mention also NIL-MP, because it is used all over the experiments.
- How many rats were used in total? In line 90, the authors should correct the age of the rats (5 weeks, rather than 0-5 weeks).
- In figure 8, the authors show interesting results in decreasing Ca2+ and calpain activity with NIL treatments. However, the data stopped at 9 weeks, based on the cataract formation (fig 7). Could the authors provide the data at 11 weeks to see if the results are similar to the none reference or if the NIL still have a longer effect but not strong enough to delay partially the cataract formation?
- Cataract can be treated with a surgery, as a routine surgery. In term of translational medicine, the authors should discuss if delaying the cataract formation will improve the patients sight, or have an impact in cataract treatments and explain what is the advantage of such treatment compare to a surgery?
Sincerely
Author Response
We carefully revised our manuscript according to the suggestions of the reviewer 4, and details are as follows.
< Q and A for Reviewer 4>
Q1. In the abstract, the authors should mention also NIL-MP, because it is used all over the experiments.
A1. The reviewer’s comment is correct. In order to respond to the reviewer’s comment, we mentioned about the NIL-MP in the abstract (Abstract).
Q2. How many rats were used in total? In line 90, the authors should correct the age of the rats (5 weeks, rather than 0-5 weeks).
A2. Thank you for pointing out this. The 162 rats were used in this study, and we corrected the age of the rats to “5 weeks” (line 92).
Q3. In figure 8, the authors show interesting results in decreasing Ca2+ and calpain activity with NIL treatments. However, the data stopped at 9 weeks, based on the cataract formation (fig 7). Could the authors provide the data at 11 weeks to see if the results are similar to the none reference or if the NIL still have a longer effect but not strong enough to delay partially the cataract formation?
A3. The reviewer’s comments are very important. At 11 weeks of age, the Ca2+ and calpain activity between SCRs instilled with NIL-NP dispersions (Ca2+ content 15.6±0.18 µmol/mg wet weight, calpain activity 187±16.1%, n=8) were significant lower than that in non-instilled SCRs (Ca2+ content 2.13±0.19 µmol/mg wet weight, calpain activity 231±10.8%, n=8). This result show that the NIL have a longer effect but not strong enough to delay partially the cataract formation. In order to respond to the reviewer’s comment, we added the content in the Discussion (line 478-483).
Q4. Cataract can be treated with a surgery, as a routine surgery. In term of translational medicine, the authors should discuss if delaying the cataract formation will improve the patients sight, or have an impact in cataract treatments and explain what is the advantage of such treatment compare to a surgery?
A4. Thank you very much for pointing this out. The cataracts can be treated with the surgical replacement of the opacified lens, however, this surgery is not easily performed in developing countries. Therefore, alternative treatments as well as treatment strategies using topical drugs are required to treat patients facing the threat of blindness. From these background, we think that the delay of cataract formation will improve the patients sight. In addition, we previously reported that the instillation of disulfiram (radical scavenger) and lanosterol (solubilization of crystallin aggregation) were also prevent the onset of cataract. The combination of NIL and other anti-cataract drugs with different preventive mechanisms may be effective in treatment of cataracts. The combination may provide an impact in cataract treatments. In order to respond to the reviewer’s comment, we added these contents in the Discussion (line 487-493, References 46 and 47).
Thank you for great comments.

Round 2
Reviewer 3 Report
I have carefully checked the authors’ responses to my comments. The revised version has adequately addressed all the critiques raised by this reviewer and is now suitable for publication in a high-quality journal "Pharmaceutics".
Reviewer 4 Report
Dear authors,
The authors revised the manuscript accordingly to the questions requested by the reviewers. I have no additional comments.